# Estimating causal effects of physical disability and number of comorbid chronic diseases on risk of depressive symptoms in an elderly Chinese population: a machine learning analysis of cross-sectional baseline data from the China longitudinal ageing social survey

Zhenjie Wang [1], Hanmo Yang [2], Chenxi Sun,[3] Shenda Hong [4]

For numbered affiliations see end of article.

**Correspondence to**
Dr Zhenjie Wang;
zhenjie.wang@pku.edu.cn and
Dr Shenda Hong;
hongshenda@pku.edu.cn

## ABSTRACT

**Objective** This study aimed to explore the causal effects of physical disability and number of comorbid chronic diseases on depressive symptoms in an elderly Chinese population.

**Design, setting and analysis** Cross-sectional, baseline data were obtained from the China Longitudinal Ageing Social Survey, a stratified, multistage, probabilistic sampling survey conducted in 2014 that covers 28 of 31 provincial areas in China. The causal effects of physical disability and number of comorbid chronic diseases on depressive symptoms were analysed using the conditional average treatment effect method of machine learning. The causal effects model's adjustment was made for age, gender, residence, marital status, educational level, ethnicity, wealth quantile and other factors.

**Outcome** Assessment of the causal effects of physical disability and number of comorbid chronic diseases on depressive symptoms.

**Participants** 7496 subjects who were 60 years of age or older and who answered the questions on depressive symptoms and other independent variables of interest in a survey conducted in 2014 were included in this study.

**Results** Physical disability and number of comorbid chronic diseases had causal effects on depressive symptoms. Among the subjects who had one or more functional limitations, the probability of depressive symptoms increased by 22% (95% CI 19% to 24%). For the subjects who had one chronic disease and those who had two or more chronic diseases, the possibility of depressive symptoms increased by 13% (95% CI 10% to 15%) and 20% (95% CI 18% to 22%), respectively.

**Conclusion** This study provides evidence that the presence of one or more functional limitations affects the occurrence of depressive symptoms among elderly people. The findings of our study are of value in developing programmes that are designed to identify elderly individuals who have physical disabilities or comorbid chronic diseases to provide early intervention.

### STRENGTHS AND LIMITATIONS OF THIS STUDY

⇒ The current study used a large, representative population-based sample covering 28 of 31 provincial areas in China.
⇒ The study estimated causal effects using the conditional average treatment effect method.
⇒ We were unable to evaluate the effects of lifestyle factors since information on these factors was not collected in the China Longitudinal Ageing Social Survey.
⇒ The findings of this cross-sectional study established a cause-and-effect relationship based on the use of a machine learning algorithm; these must be considered in further research.

### INTRODUCTION

Currently, depressive symptoms (DSs) are one of the most prevalent mental disorders worldwide.[1] The characteristics of DS include symptoms of sadness, depressed mood and loss of interest.[1] The presence of DS is usually associated with low quality of life, cancer, chronic diseases, suicide and other conditions.[2–5] DS can also place a heavy burden on families, communities and health services worldwide.[6 7] The prevalence of DS varies widely and ranges from 1% to 16% among middle-aged and elderly populations.[8–10] In China, the prevalence of DS among elderly people has increased rapidly from 3.9% to 17% in the past two decades.[11–13]

Many studies have provided evidence that specific demographic variables are associated with DS risk,[14] although the associations are inconsistent.[10 13] An inverse or U-shaped association between age and DS has been

found in many studies.[15–17] Demographic status and level of education were also found to be associated with DS risk.[18–20] However, some studies reported no association of DS with demographic status or level of education.[14 17 21]

Machine learning (ML) models can handle complex relationships between variables and can capture non-linearities and interactions that may be missed by traditional statistical models. Currently, researcher can evaluate the treatment effect though the average effect treatment effect.[22] Conditional average treatment effect (CATE) estimation is a specific ML technique used in the context of causal inference. CATE aims to estimate the effect of a treatment on an outcome for a specific subpopulation or individual, rather than the average effect across the entire population. Very few studies have used ML to obtain evidence regarding the causal effects of physical disability or the number of comorbid chronic diseases on DS risk in the elderly Chinese population.

In this study, we used ML to explore the CATE of physical disability and number of comorbid chronic diseases on DS using data from the Chinese Longitudinal Ageing Social Survey (CLASS).

## METHODS
### Data source
We drew the sample used in the current study from the cross-sectional baseline data the CLASS, which was conducted in 2014.[23] The CLASS was collected by the National Survey Research Center, Renmin University. The CLASS used a stratified, multistage, probabilistic sampling method to select a nationally representative sample covering 28 of 31 provincial areas in China. A total of 11 511 older adults were surveyed. In this study, the sample comprised 7496 subjects 60 years of age or older who answered the questions on DS and other independent variables of interest collected in 2014. All the participants were interviewed face to face by trained interviewers.

### Measurement of variables
#### Depressive symptoms
DSs were assessed using the nine-item Centre for Epidemiological Studies Depression Scale (CES-D). This scale includes three items designed to assess positive feelings and two items each designed to assess negative emotions, somatic symptoms and sense of marginalisation. The nine-item CES-D is reliable and valid for detecting nonpsychotic mental disorders among older Chinese adults.[24] For each item, a score of 0 (rarely or none of the time), 1 (some of the time) or 2 (most of the time) was assigned; thus, the total score ranged from 0 to 18. Since the coding of the positive effect items was reversed, a higher score indicates a higher level of DSs. In the current study, on the nine-item scale, the total possible score was 18 (9 items multiplied by 2, the highest response). A standardised cut-off score of 4.8 for the nine-item form of the CES-D

was established using Kohout's formula.[25] In this study, the internal Cronbach's alpha for the nine items was 0.75.

### Physical disabilities
Activities of daily living (ADLs) were assessed using the Barthel Index. For each individual, the self-reported number of comorbid chronic diseases that were based on self-reported previous medical record, including health problems such as hypertension, diabetes, heart disease, renal disease, liver disease, stroke, tuberculosis, arthritis and respiratory diseases, was categorised as '0', '1' or '≥2'.

### Confounding variables
Confounding variables, which were significantly associated with DSs,[26] under consideration were gender (male, female), age, marital status (married, widowed/divorced/unmarried), ethnicity (Han, others), residence (rural, urban), educational level (junior high school and above, primary school, never attended school) and living arrangements (lives alone, lives with others). Income was categorised into five levels using quintiles of household income (yuan) (Q1: ≤3000; Q2: >3000 and ≤10 000; Q3: >10 000 and ≤24 000; Q4: >24 000 and ≤36 000; Q5: >36 000).

### Estimation of causal effects
Causal analysis aims to determine the causal relationships between variables (treatments) and outcomes. Causal relationships are essentially different from correlations in that causal relationships mainly solve the problem of 'why'. In this work, we used causal ML, an open-source Python software provided by Uber, to perform causal analysis using ML algorithms based on recent research.[27] Causal ML provides a standard interface that allows users to estimate the average treatment effect (ATE) based on experimental or observational data.

To estimate the causal relationship between the variable of interest (known as treatment indicator E) and the outcome Y, we performed a two-step analysis: (1) we built a ML model and (2) we estimated the ATE based on causal ML. First, we built a random forest classifier (RFC) and used it to predict DS using all the variables except the treatment indicator (denoted X).[28] The RFC contains 100 decision trees and uses the Gini purity index to judge the segmentation criteria of a tree node. The maximum depth of a single tree was set to unlimited so that each tree would be segmented until each leaf node of the tree model contained only one sample. The RFC output was 1 if the individual had DS and 0 otherwise. The RFC model was built using a scikit-learn ML framework.[29]

We then set the above trained RFC as the base learner for causal ML and estimated the treatment effect based on the algorithm described here.[30] The algorithm, which uses a single base learner from the ML model and has the treatment indicator as a feature, operates in two stages. In the first stage, the algorithm estimates the average outcomes after specific treatments using the trained RFC model, where the outcome is calculated as $\mu(x) = E(Y|X = x, E = e)$. In the second stage, the ATE

**Table 1** Subjects' characteristics by their depressive symptom status according to the 9-item Centre for Epidemiological Studies Depression Scale among the elderly population in China

| Characteristics | Lower than depressive symptom cut-off point* | Higher than depressive symptom cut-off point* | P value |
|---|---|---|---|
| Age (years) | | | |
| 60–64 | 1603 (37.2) | 1068 (33.5) | <0.001 |
| 65–69 | 988 (22.9) | 704 (22.1) | |
| 70–74 | 766 (17.8) | 547 (17.2) | |
| 75–79 | 530 (12.3) | 471 (14.8) | |
| 80+ | 422 (9.8) | 397 (12.5) | |
| Residence | | | |
| Rural | 1207 (28.0) | 1315 (41.3) | <0.001 |
| Urban | 3102 (72.0) | 1872 (58.8) | |
| Gender | | | |
| Male | 2427 (56.3) | 1627 (51.1) | <0.001 |
| Female | 1882 (43.7) | 1563 (48.9) | |
| Marital status | | | |
| Married | 3316 (77.0) | 2038 (63.9) | <0.001 |
| Widowed/divorced/unmarried | 993 (23.0) | 1149 (36.1) | |
| Educational level | | | |
| Never attended school | 651 (15.1) | 816 (25.6) | <0.001 |
| Primary school | 1409 (32.7) | 1266 (39.7) | |
| Junior high school and above | 2249 (52.2) | 1105 (34.7) | |
| Ethnicity | | | |
| Han | 4074 (94.5) | 2990 (93.8) | 0.19 |
| Other | 235 (5.5) | 197 (6.2) | |
| Living arrangement | | | |
| Live with others | 3925 (91.1) | 2666 (83.7) | <0.001 |
| Live alone | 384 (8.9) | 521 (16.3) | |
| Physical disability | | | |
| No functional limitations | 3988 (92.6) | 2531 (79.4) | <0.001 |
| One or more functional limitations | 321 (7.4) | 656 (20.6) | |
| Wealth quantile | | | |
| Q1 (lowest) | 691 (16.0) | 925 (29.0) | <0.001 |
| Q2 | 694 (16.1) | 717 (22.5) | |
| Q3 | 1066 (24.7) | 705 (22.1) | |
| Q4 | 1002 (23.3) | 520 (16.3) | |
| Q5 (highest) | 856 (19.9) | 320 (10.1) | |
| No of comorbid chronic diseases | | | |
| 0 | 1463 (34.0) | 551 (17.3) | <0.001 |
| 1 | 1307 (30.3) | 917 (28.8) | |
| ≥2 | 1539 (35.7) | 1719 (53.9) | |

*For the current study, on a nine-item scale, the total possible score was 18 (9 items multiplied by 2, the highest response). The standardised cut-off point was 4.8 by using Kohout's formula.

was calculated as $\tau(x) = \mu(x, E = 1) - \mu(x, E = 0)$. We also used SHAPley Additive exPlanations, a game theory approach, to explain the output of the causal analysis results modelled by the ML models.[31] We analysed the data using Python V.3.7.14 based on the anaconda3 development platform, causalml V.0.10.0, sklearn V.0.23.2, pandas V.1.0.1 and numpy V.1.21.0.

**Survey conduct and consent**

The interviewer-administered questionnaire was prepared in the local language (Chinese). Consent to participation

**Table 2** Estimated causal effects of physical disability and chronic disease on depressive symptoms

| | Reference | | Causal effect (95% CI) |
|---|---|---|---|
| Physical disability* | No functional problems | One or more functional limitations | 0.22 (0.19 to 0.24) |
| No of comorbid chronic diseases† | 0 | 1 | 0.13 (0.10 to 0.15) |
| | | ≥2 | 0.20 (0.18 to 0.22) |

*Taking into consideration, the causal effects of age, gender, residence, marital status, educational level, ethnicity, wealth quantile and number of comorbid chronic diseases.
†Taking into consideration, the causal effects of age, gender, residence, marital status, educational level, ethnicity, wealth quantile and physical disability.

in the study was received from each participant prior to data collection, and the data collectors were trained to provide any requested information or clarification at any time during the interview. Only participants who wished to continue participating in the study after providing informed consent were included in the study.

## Patient and public involvement
None.

## RESULTS
### Descriptive statistics
The characteristics of subjects with and without DSs are summarised in table 1. The overall average CES-D score was 4.56 (SD: 3.56). The prevalence of DS in the Chinese population over 60 years of age was 43%. In the current study, male subjects, urban residents, people living with others and people of Han ethnicity accounted for the majority of elderly Chinese individuals with DS.

### Estimated causal effects
The estimated causal effects of physical disability and chronic diseases on DSs are presented in table 2. Physical disability and the number of comorbid chronic diseases had causal effects on DS. Among the subjects who had one or more functional limitations, those who had one chronic disease, and those who had two or more chronic diseases, the risk probability of DS increased by 22%, 13% and 20%, respectively.

The effects of the mechanism direction of physical disability and number of comorbid chronic diseases on DS are presented in figures 1 and 2 in descending order of importance In descending order of importance among subjects who had one or more functional limitations, the variables are as follows: educational level, age, number of comorbid chronic diseases, wealth quantile, marital status, living arrangement, residence, gender and ethnicity. In descending order of importance among subjects who had only one chronic disease, the variables are as follows: wealth quantile, age, educational level, residence, gender, marital status, physical disability, living arrangement and ethnicity. In descending order of importance among subjects who had one or more functional limitations, the variables are as follows: wealth quantile, age, residence, educational level, gender, marital status, living arrangement, physical disability and ethnicity. Age, residence, gender and physical disability contributed to DS risk among subjects who had one or more chronic diseases.

## DISCUSSION
Overall, physical disability and the number of comorbid chronic diseases present had causal effects on DS risk. Subjects who had one or more functional limitations had a 22% increased risk of DS. Subjects who had one chronic

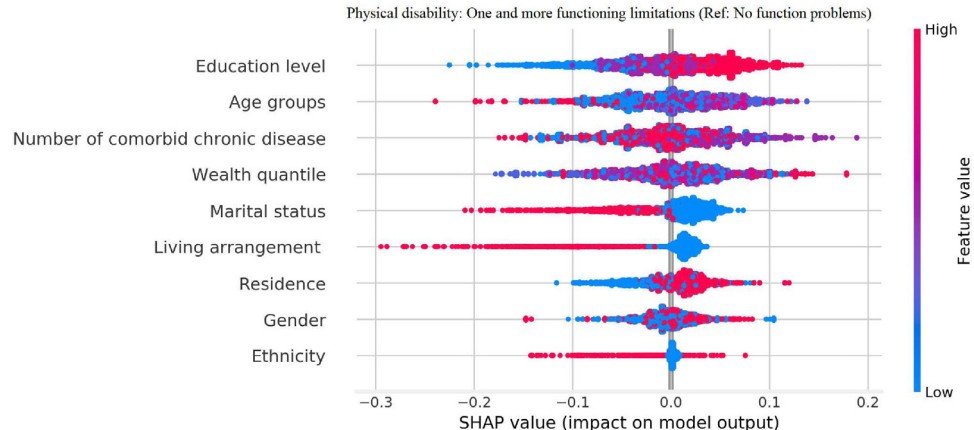

**Figure 1** The importance sort and mechanism direction of physical disability with depressive symptoms by using explainable machine learning predictions. The importance of variables (Mean of SHAP value with 3 decimal places: risk factor (RF): mean of SHAP value >0; protective factor (PF): mean of SHAP value <0; no effect (NE): mean of SHAP value=0) by descending were education level (NE), age groups (RF), number of comorbid chronic disease (PF), wealth quantile (NE), marital status (PF), living arrangement (NE), residence (RF), gender (RF) and ethnicity (PF). SHAP, SHAPley Additive exPlanations.

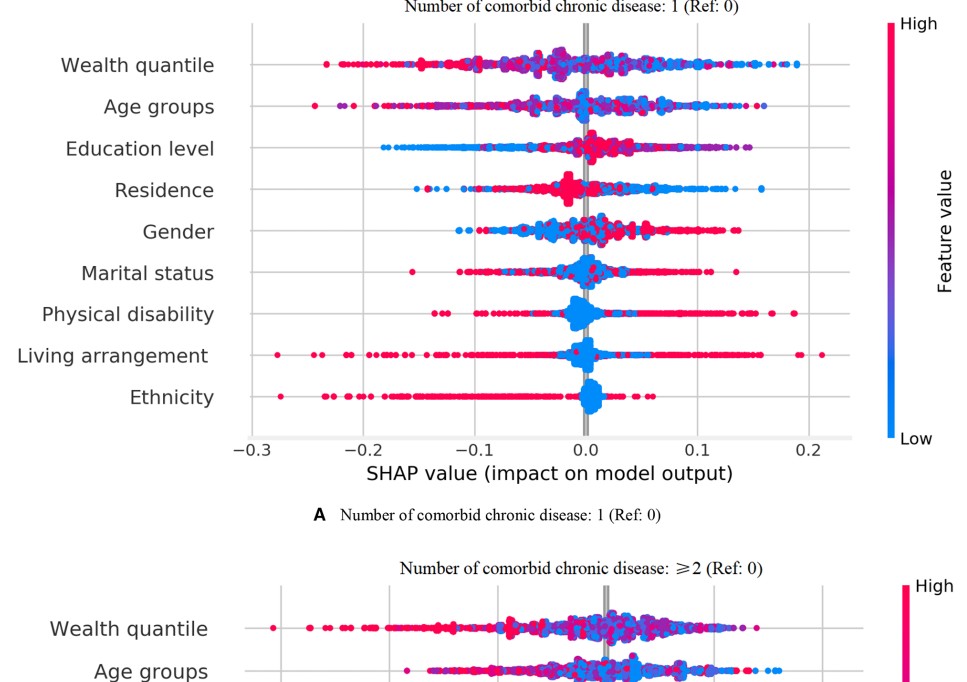

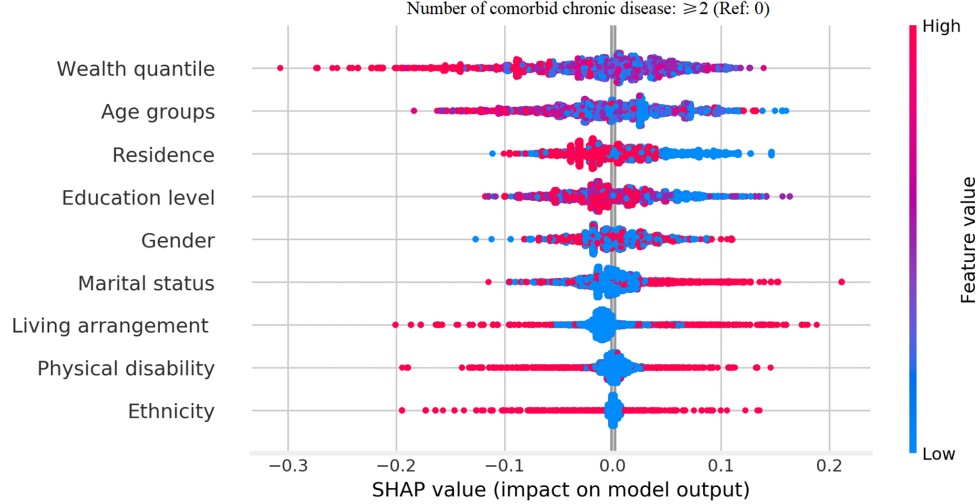

**Figure 2** The importance sort and mechanism direction of number of comorbid chronic disease with depressive symptoms by using explainable machine learning predictions. (A) Number of comorbid chronic disease: 1 (Ref: 0). The importance of variables (mean of SHAP value with 3 decimal places: risk factor (RF): mean of SHAP value >0; protective factor (PF): mean of SHAP value <0; no effect (NE): mean of SHAP value=0) by descending were wealth quantile (NE), age groups (RF), education level (RF), residence (RF), gender (RF), marital status (PF), physical disability (RF), living arrangement (NE) and ethnicity (PR). (B) Number of comorbid chronic disease: ≥2 (Ref: 0). The importance of variables (mean of SHAP value with 3 decimal places: risk factor (RF): mean of SHAP value >0; protective factor (PF): mean of SHAP value <0; no effect (NE): mean of SHAP value=0) by descending were wealth quantile (PF), age groups (RF), residence (RF), education level (NE), gender (RF), marital status (NF), living arrangement (NE), physical disability (NF) and ethnicity (PF). SHAP, SHAPley Additive exPlanations.

disease and those who had two or more chronic diseases showed increases of 13% and 20%, respectively, in their DS risk.

In the current study, we found causal effects of physical disability and comorbid chronic diseases on DS. A previous study suggested that some health variables, such as poor self-rated health and disability, are associated with DS, indicating that health status tends to have similar effects on mental health regardless of social context.[32] Physical disability is considered an inability or a diminished capacity to perform basic self-care activities. Physical disability lessens an individual's ability to interact with the physical and social environment,

hence increasing the risk of DS.[33] Increased disability in ADLs and instrumental ADLs have been proven to be connected with DSs.[34 35] Failure to perform ADLs has been positively associated with DSs in elderly people, and physical disability accounts for incidences of DSs in elderly individuals.[36 37] Furthermore, the finding that physical disability leads to an increase in DSs in geriatric individuals over time is supported by longitudinal studies.[38] Given the influence of physical disability and DSs on the well-being of elderly individuals, it is urgent to find ways to address these issues. The results presented here demonstrate that impairment of ADLs has a significant effect on DSs in elderly people, consistent with the

findings of previous studies that showed a relationship between limitations on activity and psychosocial problems.[33 34]

The number of chronic diseases present in an individual was found to be associated with DS risk, consistent with previous studies. These associations could be explained by several factors, including unhealthy lifestyle, shared genetic influences and inflammation related to disease.[39] Previous studies have suggested that experiencing pain and physical disability could affect mood in elderly individuals.[20] Another explanation is that subjects with DSs might be more sedentary than those without DSs; such a lifestyle could cause increased bone resorption and decreased bone formation.[40 41]

Causal inference using ML methods such as CATE can be a powerful tool for analysing observational data. They model the treatment effect using regression or decision trees. ML can learn causal effects because it can identify and account for confounding factors, which are variables that may affect both the treatment and outcome variables. By controlling for confounding factors, ML algorithms can estimate the causal effect of the treatment variable on the outcome variable, even in the absence of a randomised controlled trial. Specifically, CATE aims to estimate the effect of a treatment on an outcome for a specific subpopulation or individual, rather than the average effect across the entire population. This method can help identify treatment effects for different subgroups of the population and enable personalised treatment recommendations.

## Strengths and limitations

Our study has many strengths, including a large sample size, a population-based design and adjustment for a wide range of socioeconomic characteristics. Another noticeable strength is that the measurement of all physical illnesses took place prior to the CES-D measurement, minimising the risk of reverse causation. However, our study also has several limitations that should be taken into account by future researchers. First, the physical conditions information were self-reported, which should also be considered for future research. Moreover, although we used ML to estimate the causal effects of the factors on DS, these findings should be interpreted cautiously in view of the cross-sectional nature of the data used in the present analysis. We will be able to extend the current study to determine causality when longitudinal data are available. Although the current data were collected in 2014, they are available on the web. Because all the information was collected by trained interviewers, the quality of the survey was reliable. This point should also be considered with caution by future researchers. It is important to note that the CLASS does not provide information on lifestyle factors (ie, weight, height, smoking, alcohol consumption and other factors), some of which have been suggested as risk factors for depression.[42–44]

## CONCLUSION

In conclusion, in this large population-based study in an elderly Chinese population, physical disability and the number of comorbid chronic diseases were found to increase DS risk. This study provides evidence of the effect of one or more functional limitations on the occurrence of DSs among elderly people. This is helpful in planning preventive measures and in improving knowledge of how to treat DSs. The findings made in our study are valuable for the development of prevention programmes designed to identify elderly individuals who have physical disabilities or a number of comorbid chronic diseases and provide early intervention.

**Author affiliations**
[1]Institute of Population Research, Peking University, Beijing, People's Republic of China
[2]T. H. Chan School of Public Health, Harvard University, Boston, Massachusetts, USA
[3]School of Intelligence Science and Technology and the Key Laboratory of Machine Perception (Ministry of Education), Peking University, Beijing, People's Republic of China
[4]National Institute of Health Data Science, Peking University, Beijing, People's Republic of China

**Acknowledgements**  We would like to thank the Institute of Gerontology and National Survey Research Center at Renmin University of China for providing the CLASS data.

**Contributors**  Conceptualisation: ZW; methodology: SH; software: ZW and SH; writing: ZW; writing—review: ZW, SH, HY and CS; writing—editing: HY and CS.

**Funding**  This work was supported by the National Social Science Fund of China (No. 20BRK020), the National Natural Science Foundation of China (No. 62102008) and the Economic & Social Research Council (ESRC) (No. ES/P011055/1).

**Disclaimer**  The funding agencies did not participate in the research or review any details of this study, and the authors are independent of the funder

**Competing interests**  None declared.

**Patient and public involvement**  Patients and/or the public were not involved in the design, or conduct, or reporting, or dissemination plans of this research.

**Patient consent for publication**  Consent obtained directly from patient(s).

**Ethics approval**  This study involves human participants but all procedures performed in this study involving human participants were in accordance with the ethical standards of the institutional and/or national research committee and with the 1964 Declaration of Helsink and its later amendments or comparable ethical standards. The surveys were conducted within the legal framework governed by statistical law in China. This article does not contain any studies with animals performed by any of the authors. Participants gave informed consent to participate in the study before taking part.

**Provenance and peer review**  Not commissioned; externally peer reviewed.

**Data availability statement**  Data are available on reasonable request. The datasets generated and/or analysed during the current study are available atin the China Longitudinal Ageing Social Survey (CLASS) repository, (http://class.ruc.edu.cn/index.php?r=index/index&hl=en).

**ORCID iDs**
Zhenjie Wang http://orcid.org/0000-0003-1717-5790
Hanmo Yang http://orcid.org/0000-0003-4848-0878
Shenda Hong http://orcid.org/0000-0001-7521-5127

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
