## [Reviewer comments · BMJ Open]

ARTICLE DETAILS

TITLE (PROVISIONAL)	Estimating causal effects of physical disability and number of comorbid chronic diseases on risk of depressive symptoms in an elderly Chinese population: a machine learning analysis of cross-sectional baseline data from the China Longitudinal Ageing Social Survey
AUTHORS	Wang, Zhenjie; Yang, Hanmo; Sun, Chenxi; Hong, Shenda

VERSION 1 – REVIEW

REVIEWER	Claudia Iveth Astudillo-García Instituto Nacional de Psiquiatría Ramón de la Fuente Muñiz
REVIEW RETURNED	12-Dec-2022

GENERAL COMMENTS	This manuscript is aimed to explore the causal effects of physical disability and number of comorbid chronic diseases on depressive symptoms in an elderly Chinese population. Although they have a representative sample, which adds to the heterogeneous evidence reported on depressive symptom in elderly population, I am concerned that the data, being derived from a cross-sectional study, do not provide evidence for causal effects, even when using sophisticated machine learning methods. To consider causal effects between the exposure variables evaluated (physical disability and chronic diseases) and depressive symptoms, the latter should be taken from cohorts where the outcome was not present, so that we could speak of incident depression. While causal Machine Learning provides a standard interface that allows users to estimate the average treatment effect from experimental or observational data, this data are observational. Therefore, it is important to note that despite the causal models used, the data only allow for predictions rather than causal explanations. While the authors can decide between the best models to achieve their objectives, the theory and methodology should determine this decision. I suggest then that the authors discuss the scope of their results and focus on predictive models of causal models. Moreover, the authors already describe the results in terms of possibilities and not risk, which is evidence of the correlational and not causal scope of the study. Therefore, I recommend not focusing on the cause-effect relationship, but to soften or change the term, either to prediction using causal models or some other that they consider prudent. I believe that the analysis employed can be a strength in adding evidence of the relationship between risk factors and outcome, rather than positioning it as the overall contribution of the study.
--

	By other hand, I have others comments that are described below by section: Abstract:  - I recommend not repeating the sample size data and ages of participants in the results section (they point this out in the participants section as well) - Add the description of the method, noting the design used and the analyses performed (unless the journal guidelines indicate otherwise) - Add the period in which each condition was evaluated, exposures and outcomes. Introduction Consider restructuring the objective. Method Data source  - Insert the reference to the Chinese Longitudinal Ageing Social study. - Add at what time each condition was assessed Measurement of variables  - Add the study design type Depressive symptoms  - Describe whether the depression that was assessed was incident depression, which should have ensured that the subjects in the analysis did not present it at time 0 Sociodemographic characteristics  - Describe how Wealth quantile was assessed - Describe whether measures of physical disability and chronic illness were self-reported Estimating causal effects  - Add references in the sentence: open source Python software provided by Uber, for causal analysis methods using machine learning algorithms based on recent research Discussion  - Discuss elements of the internal validity of the study: quality of measurements, whether they were self-reported, etc. Limitations  - Add whether the data would still be current despite the time of collection.
--	---

REVIEWER	Yin Kejing Hong Kong Baptist University
REVIEW RETURNED	18-Jan-2023

GENERAL COMMENTS	The manuscript examines the causal effects of demographic factors and physical disabilities on depressive symptoms (DS) in older adults in China. Overall, the paper is well organized and the methods used are reasonable. I only have a few minor comments/suggestions:  1. Page 6, lines 56-59: This sentence seems to contain grammar mistakes, please double check. 2. Page 7, line 30: "consisted" should be "is consisted". 3. In the response to previous reviews, the authors conduct further experiments and measured the ROC-AUC scores with and without the features of interest. It is suggested to add the metrics to the paper as readers may wonder the same.
--

REVIEWER	Michiaki Higashitani Tokyo Medical University Ibaraki Medical Center, Cardiology
REVIEW RETURNED	11-Feb-2023

GENERAL COMMENTS	I find your paper interesting. It is reasonable that physical disabilities and comorbid chronic diseases affect depressive symptoms (DS), as reported previously. Major findings) 1) The evaluation method of estimating physical disability is not noted. For example, Barthel Index or Katz Index is listed. Moreover, every score distribution of the evaluation method should be listed. 2) This study would be more interesting to see how physical disability specifically affects DS. You have to reveal how every score distribution of physical disability affects DS. If you can't show it, you should state why. 3) Is the number of comorbid chronic diseases correlated with the onset of DS, that is, the higher the number, the higher the risk of developing DS? Otherwise, is there a flat relationship in two or more chronic comorbidities? 4) Is there a difference in the impact on DS depending on the type of comorbid chronic diseases? whether there are some comorbid chronic diseases, including age and gender, that have a large impact on the onset of DS, and whether they differ according to risk. 5) I don't understand the meaning of two Figures. Does the difference between the red and blue plots mean the high and low of the Feature Value on vertical axis on the right side of the figure? It is unclear what the order from the top of subjects such as Education level means. Once again, please explain the details so that I can understand the meaning of the figure.
---

REVIEWER	Xiaowei Dong University of Gothenburg
REVIEW RETURNED	20-Feb-2023

GENERAL COMMENTS	Review of bmjopen-2022-069298 Estimating causal effects of physical disability and number of comorbid chronic diseases on depressive symptom risk in an elderly Chinese population: A machine learning study Comments to the author The study estimated the causal effects based on a good representative sample using conditional average treatment effect method, which are the main strengths of the paper. However, there are some concerns relating to the manuscript that needs to be developed. Introduction The manuscript mentioned the inconsistent relationships between a variety of demographic factors and depressive symptoms. Since the aim of the study was to estimate the causal effects of physical disability and number of comorbid chronic diseases, it may be more appropriate to present the noncausal association of those two exposure variables with depressive symptoms in the previous literature. Methods and materials Please elaborate on how potential confounders were selected. The exposures of this study (i.e., physical disability and comorbid chronic diseases) may need to be differentiated from demographic factors.
--

	The descriptions of sociodemographic characteristics seem not include wealth quantile, while this variable was included in table 1. Please state the assumptions required for causal inference. Are these assumptions satisfied? Results The authors stated that “the possibility of DS increased by 22%, 13%, and 20%, respectively”. However, physical disability and number of comorbid chronic diseases have different reference levels. It is recommended to report the causal estimates of two variables separately, stating which reference they compared to. The authors used the term “possibility of DS”. Is this risk ratio or odds ratio? The term needs to be used consistently throughout the manuscript. Tables and figures In table 1, the characteristics of respondents were presented according to the following two groups: without depressive symptom and with depressive symptom. However, individuals with a CES-D score below the cut-off point may also have some depressive symptoms (if the score does not equal to zero). Could this be worded differently?
--	--

VERSION 1 – AUTHOR RESPONSE

Response to Reviewer 1

Dear Reviewer,

We are extremely grateful for your review of the manuscript. You have raised a number of important issues. We agree with your comments and have modified our manuscript accordingly, as documented below.

Reviewer's report

This manuscript is aimed to explore the causal effects of physical disability and number of comorbid chronic diseases on depressive symptoms in an elderly Chinese population. Although they have a representative sample, which adds to the heterogeneous evidence reported on depressive symptom in elderly population, I am concerned that the data, being derived from a cross-sectional study, do not provide evidence for causal effects, even when using sophisticated machine learning methods. To consider causal effects between the exposure variables evaluated (physical disability and chronic diseases) and depressive symptoms, the latter should be taken from cohorts where the outcome was not present, so that we could speak of incident depression.

While causal Machine Learning provides a standard interface that allows users to estimate the average treatment effect from experimental or observational data, this data are observational. Therefore, it is important to note that despite the causal models used, the data only allow for predictions rather than causal explanations.

While the authors can decide between the best models to achieve their objectives, the theory and methodology should determine this decision. I suggest then that the authors discuss the scope of their results and focus on predictive models of causal models. Moreover, the authors already describe the results in terms of possibilities and not risk, which is evidence of the correlational and not causal scope of the study.

Therefore, I recommend not focusing on the cause-effect relationship, but to soften or change the term, either to prediction using causal models or some other that they consider prudent. I believe that the analysis employed can be a strength in adding evidence of the relationship between risk factors and outcome, rather than positioning it as the overall contribution of the study.

By other hand, I have others comments that are described below by section:

Abstract:

- I recommend not repeating the sample size data and ages of participants in the results section (they point this out in the participants section as well)
- Add the description of the method, noting the design used and the analyses performed (unless the journal guidelines indicate otherwise)
- Add the period in which each condition was evaluated, exposures and outcomes.

Response: Thank you for your comment. We deleted the “the sample the sample size data and ages of participants” in the results section.

Introduction

Consider restructuring the objective.

Response: Thank you for your comment. We restructured the objective in the “Introduction” section. P4: L16-21.

Method

Data source

- Insert the reference to the Chinese Longitudinal Ageing Social study.
- Add at what time each condition was assessed

Response: Thank you for your comment. We added the reference to the Chinese Longitudinal Ageing Social study. P5: L5. And added the time of each variables information was collected. P5: L11.

Measurement of variables

- Add the study design type

Response: Thank you for your comment. The design of current study was cross-sectional and added this into the title as editor suggested. P1: L1-4.

Depressive symptoms

- Describe whether the depression that was assessed was incident depression, which should have ensured that the subjects in the analysis did not present it at time 0

Response: Thank you for your comment. We used the cross-sectional baseline data from the China Longitudinal Ageing Social Survey. We could not assessed the incident of depression.

Sociodemographic characteristics

- Describe how Wealth quantile was assessed
- Describe whether measures of physical disability and chronic illness were self-reported

Response: Thank you for your comment. We added the information of wealth quantile in the “Sociodemographic characteristics” section. P6: L18-19 We added the self-reported information in the “Sociodemographic characteristics” section. P6: L9.

Estimating causal effects

- Add references in the sentence: open source Python software provided by Uber, for causal analysis methods using machine learning algorithms based on recent research

Response: Thank you for your comment. We added reference “Huigang Chen, Totte Harinen, Jeong-Yoon Lee, Mike Yung, and Zhenyu Zhao. "Causalm: Python package for causal machine learning." arXiv preprint arXiv:2002.11631 (2020).” into the text.

Discussion

- Discuss elements of the internal validity of the study: quality of measurements, whether they were self-reported, etc.

Response: Thank you for your comment. We discussed the comments that reviewer suggested in the "Strengths and limitations". P13: L11-13

Limitations

- Add whether the data would still be current despite the time of collection.

Response: Thank you for your comment. We added discussion of comment that reviewer suggested. P13: L11-12.

Response to Reviewer 2

Dear Reviewer,

We are extremely grateful for your review of the manuscript. You have raised a number of important issues. We agree with your comments and have modified our manuscript accordingly, as documented below.

Reviewer's report

The manuscript examines the causal effects of demographic factors and physical disabilities on depressive symptoms (DS) in older adults in China. Overall, the paper is well organized and the methods used are reasonable.

I only have a few minor comments/suggestions:

1. Page 6, lines 56-59: This sentence seems to contain grammar mistakes, please double check.

Response: Thank you for your comment. We had checked grammar mistakes by AJE.

2. Page 7, line 30: "consisted" should be "is consisted".

Response: Thank you for your comment. We corrected this error.

3. In the response to previous reviews, the authors conduct further experiments and measured the ROC-AUC scores with and without the features of interest. It is suggested to add the metrics to the paper as readers may wonder the same.

Response: Thank you for your comment. Our aim was estimating the causal effect of physical disability and number of comorbid chronic diseases on depressive symptom risk. The aim of current study was not predicting depressive symptom. Therefore, ROC-AUC was not suitable for our aim.

Response to Reviewer 3

Dear Reviewer,

We are extremely grateful for your review of the manuscript. You have raised a number of important issues. We agree with your comments and have modified our manuscript accordingly, as documented below.

Reviewer's report

I find your paper interesting. It is reasonable that physical disabilities and comorbid chronic diseases affect depressive symptoms (DS), as reported previously.

Major findings

1) The evaluation method of estimating physical disability is not noted. For example, Barthel Index or Katz Index is listed. Moreover, every score distribution of the evaluation method should be listed.

Response: Thank you for your comment. We revised this sentence for assessing physical disability. P6: L8

2) This study would be more interesting to see how physical disability specifically affects DS. You have to reveal how every score distribution of physical disability affects DS. If you can't show it, you should state why.

Response: Thank you for your comment. It was very hard for every score distribution of physical disability affects DS in the current study. We added this some description on this point in the "limitation".

3) Is the number of comorbid chronic diseases correlated with the onset of DS, that is, the higher the number, the higher the risk of developing DS? Otherwise, is there a flat relationship in two or more chronic comorbidities?

Response: Thank you for your comment. The number greater than zero was meaning increase risk. The higher number of comorbid chronic diseases increased the risk possibility (0.13, 95%CI: 0.10-0.15), (0.20, 95%CI: 0.18-0.22) of DS, respectively. (Table 2)

4) Is there a difference in the impact on DS depending on the type of comorbid chronic diseases? whether there are some comorbid chronic diseases, including age and gender, that have a large impact on the onset of DS, and whether they differ according to risk.

Response: Thank you for your comment. The risk possibility of DS was adjusted by age group, gender, residence, marital status, education level, ethnicity, wealth quantile, wealth quantile and physical disability. We guessed that we had considered age and gender's impact on the DS in our model.

5) I don't understand the meaning of two Figures. Does the difference between the red and blue plots mean the high and low of the Feature Value on vertical axis on the right side of the figure? It is unclear what the order from the top of subjects such as Education level means. Once again, please explain the details so that I can understand the meaning of the figure.

Response: Thank you for your comment. The red and blue dots indicate the size of a feature value of the sample, as shown in the bar on the right; The position of the red/blue point on the coordinate axis indicates the size of its SHAP value, which is independent of color and the order of the subject; The position of a variable's point on the y-axis is meaningless, just to indicate the distribution density of the data.

Response to Reviewer 4

Dear Reviewer,

We are extremely grateful for your review of the manuscript. You have raised a number of important issues. We agree with your comments and have modified our manuscript accordingly, as documented below.

Reviewer's report

The study estimated the causal effects based on a good representative sample using conditional average treatment effect method, which are the main strengths of the paper. However, there are some concerns relating to the manuscript that needs to be developed.

Introduction

The manuscript mentioned the inconsistent relationships between a variety of demographic factors and depressive symptoms. Since the aim of the study was to estimate the causal effects of physical disability and number of comorbid chronic diseases, it may be more appropriate to present the noncausal association of those two exposure variables with depressive symptoms in the previous literature.

Response: Thank you for your comment. We revised the aim of the study in the "Introduction" section. P5: L16-17.

Methods and materials

Please elaborate on how potential confounders were selected.

The exposures of this study (i.e., physical disability and comorbid chronic diseases) may need to be differentiated from demographic factors.

The descriptions of sociodemographic characteristics seem not include wealth quantile, while this variable was included in table 1.

Please state the assumptions required for causal inference. Are these assumptions satisfied?

Response: Thank you for your comment. Potential confounders were selected based on the previous evidence, that was indicated in the "Introduction". We created a new subtitle "confounding variables" in the "Measurement of Variables" We added the information of wealth quantile in the "Sociodemographic characteristics" section. P6: L17-19

Results

The authors stated that "the possibility of DS increased by 22%, 13%, and 20%, respectively". However, physical disability and number of comorbid chronic diseases have different reference levels. It is recommended to report the causal estimates of two variables separately, stating which reference they compared to.

The authors used the term "possibility of DS". Is this risk ratio or odds ratio? The term needs to be used consistently throughout the manuscript.

Response: Thank you for your comment.

Tables and figures

In table 1, the characteristics of respondents were presented according to the following two groups: without depressive symptom and with depressive symptom. However, individuals with a CES-D score below the cut-off point may also have some depressive symptoms (if the score does not equal to zero). Could this be worded differently?

Response: Thank you for your comment. We used another word for describing depressive symptoms. “Lower than depressive symptom cut-off point” or “Higher than depressive symptom cut-off point” instead of “Without depressive symptom” or “With depressive symptom”

VERSION 2 – REVIEW

REVIEWER	Claudia Iveth Astudillo-García Instituto Nacional de Psiquiatría Ramón de la Fuente Muñiz
REVIEW RETURNED	23-Apr-2023

GENERAL COMMENTS	The topic is relevant and provides evidence on the contribution of disability on depressive symptoms with data from a representative sample of localities in China; however, I believe that the authors can contribute to the debate on whether machine learning can approach causality models in epidemiology, as it is a current topic that needs further discussion and they could contribute with their inputs. Most of all, making it clear to the reader how they construct the necessary counterfactual to determine causality. On the other hand, I share some comments by section: ABSTRACT It is of vital relevance to describe the data collection times, since in the discussion they say that "the measurement of all physical illnesses took place prior to the CES-D measurement, minimizing the risk of reverse causation". Describe the variables of adjustment that were considered. INTRODUCTION I suggest a brief description of the advantages of using machine learning and why specifically to use CATE over other options. METHODS AND MATERIALS Describe the timing of the measurement of the physical conditions, in the discussion they mention as a strength that these were evaluated before, but the authors do not give more information. Describe how the adjustment variables were taken into account in the analysis. DISCUSSION Add as a limitation the self-reporting of disability. Discuss further how this causality inference can be made from observational data. Recommend further reading: Broadbent, A., & Grote, T. (2022). Can Robots Do Epidemiology? Machine Learning, Causal Inference, and Predicting the Outcomes of Public Health Interventions. Philosophy & technology, 35(1), 14. https://doi.org/10.1007/s13347-022-00509-3 Weed D. L. (2016). Commentary: Causal inference in epidemiology: potential outcomes, pluralism and peer review. International journal of epidemiology, 45(6), 1838–1840. https://doi.org/10.1093/ije/dyw229
--

REVIEWER	Michiaki Higashitani Tokyo Medical University Ibaraki Medical Center, Cardiology
REVIEW RETURNED	29-Mar-2023

GENERAL COMMENTS	I think your response to my review was insufficient.
--

REVIEWER	Xiaowei Dong University of Gothenburg
REVIEW RETURNED	02-Apr-2023

GENERAL COMMENTS	I am pleased with the revisions. Thank the authors for the efforts.
---

VERSION 2 – AUTHOR RESPONSE

Response to Reviewer 1

Dear Reviewer,

We are extremely grateful for your review of the manuscript. You have raised a number of important issues. We agree with your comments and have modified our manuscript accordingly, as documented below.

Reviewer's report

The topic is relevant and provides evidence on the contribution of disability on depressive symptoms with data from a representative sample of localities in China; however, I believe that the authors can contribute to the debate on whether machine learning can approach causality models in epidemiology, as it is a current topic that needs further discussion and they could contribute with their inputs. Most of all, making it clear to the reader how they construct the necessary counterfactual to determine causality. On the other hand, I share some comments by section:

ABSTRACT

It is of vital relevance to describe the data collection times, since in the discussion they say that "the measurement of all physical illnesses took place prior to the CES-D measurement, minimizing the risk of reverse causation". Describe the variables of adjustment that were considered.

Response: Thank you for your comment. We added the variables of adjustment in the "Abstract". P2: L15-16

INTRODUCTION

I suggest a brief description of the advantages of using machine learning and why specifically to use CATE over other options.

Response: Thank you for your comment. We tried to add a brief description of the conditional average treatment effect (CATE) in the "Introduction". P5 L15-21

METHODS AND MATERIALS

Describe the timing of the measurement of the physical conditions, in the discussion they mention as a strength that these were evaluated before, but the authors do not give more information.

Response: Thank you for your comment. We added more information on physical conditions as following “For each individual, the self-reported number of comorbid chronic diseases that were based on self-reported previous medical record, including health problems such as hypertension, diabetes, heart disease, renal disease, liver disease, stroke, tuberculosis, arthritis, and respiratory diseases, was categorized as “0”, “1” or “≥2”.” P7 L12-13.

Describe how the adjustment variables were taken into account in the analysis.

Response: Thank you for your comment. We describe the reason of adjustment variables were taken into account in the analysis. P7 L18-19

DISCUSSION

Add as a limitation the self-reporting of disability.

Response: Thank you for your comment. We added the limitation of self-reporting of disability in the “Discussion”. P14 L15-16.

Discuss further how this causality inference can be made from observational data.

Response: Thank you for your comment. We added some discussion on this point in the “Discussion”. P13 L20-P14 L8

Recommend further reading:

Broadbent, A., & Grote, T. (2022). Can Robots Do Epidemiology? Machine Learning, Causal Inference, and Predicting the Outcomes of Public Health Interventions. *Philosophy & technology*, 35(1), 14. <https://doi.org/10.1007/s13347-022-00509-3>

Weed D. L. (2016). Commentary: Causal inference in epidemiology: potential outcomes, pluralism and peer review. *International journal of epidemiology*, 45(6), 1838–1840. <https://doi.org/10.1093/ije/dyw229> There is a lack of clarity about the study design, given that in several places you indicate the study is cross-sectional, whereas the study name includes the word ‘longitudinal’. Are you just using the cross-sectional, baseline data from a longitudinal study? Please ensure this is clearly stated in the article title, abstract, and main text Methods section.

Response: Thank you for your comment. We only used the cross-sectional, baseline data from a longitudinal study. We changed the title, abstract, and main text Methods section.

Please revise the article title to better indicate the study design and data source. Eg, "Estimating causal effects of physical disability and number of comorbid chronic diseases on risk of depressive symptoms in an elderly Chinese population: a machine learning analysis of cross-sectional baseline data from the China Longitudinal Ageing Social Survey".

Response: Thank you for your comment. I changed our manuscript title to "Estimating causal effects of physical disability and number of comorbid chronic diseases on risk of depressive symptoms in an elderly Chinese population: a machine learning analysis of cross-sectional baseline data from the China Longitudinal Ageing Social Survey" as suggested. P1

Response to Reviewer 3

Dear Reviewer,

We are extremely grateful for your review of the manuscript. You have raised a number of important issues. We agree with your comments and have modified our manuscript accordingly, as documented below.

Reviewer's report

I think your response to my review was insufficient.

Response: Thank you for your comment. We addressed all the comments mentioned as much as we can.

Response to Reviewer 4

Dear Reviewer,

We are extremely grateful for your review of the manuscript. You have raised a number of important issues. We agree with your comments and have modified our manuscript accordingly, as documented below.

Reviewer's report

I am pleased with the revisions. Thank the authors for the efforts.

Response: Thank you for your comment.

VERSION 3 – REVIEW

REVIEWER	Claudia Iveth Astudillo-García Instituto Nacional de Psiquiatría Ramón de la Fuente Muñiz
REVIEW RETURNED	09-May-2023
GENERAL COMMENTS	The authors responded to suggested comments.